# Plant-Growth-Promoting Bacteria Can Impact Zinc Uptake in *Zea mays*: An Examination of the Mechanisms of Action Using Functional Mutants of *Azospirillum brasilense*

**DOI:** 10.3390/microorganisms9051002

**Published:** 2021-05-06

**Authors:** Alexandra Bauer Housh, Mary Benoit, Stacy L. Wilder, Stephanie Scott, Garren Powell, Michael J. Schueller, Richard A. Ferrieri

**Affiliations:** 1Missouri Research Reactor Center, University of Missouri, Columbia, MO 65211, USA; afbkhn@mail.missouri.edu (A.B.H.); wildersl@missouri.edu (S.L.W.); srstt9@mail.missouri.edu (S.S.); garren.powell@mail.missouri.edu (G.P.); schuellerm@missouri.edu (M.J.S.); 2Chemistry Department, University of Missouri, Columbia, MO 652101, USA; 3Interdisciplinary Plant Group, University of Missouri, Columbia, MO 65211, USA; 4Division of Plant Sciences, University of Missouri, Columbia, MO 65211, USA; mvbenoit@mail.missouri.edu; 5Department of Biochemistry, University of Missouri, Columbia, MO 65211, USA

**Keywords:** plant-growth-promoting bacteria, zinc nutrient uptake, maize, ^65^Zn and ^59^Fe radiotracers

## Abstract

Among the PGPB, the genus *Azospirillum*—with an emphasis on *A. brasilense*—is likely the most studied microorganism for mitigation of plant stress. Here, we report the investigation of functional mutants HM053, *ipdC* and FP10 of *A. brasilense* to understand how the biological functions of these microorganisms can affect host Zn uptake. HM053 is a *Nif*
^+^ constitutively expressed strain that hyper-fixes N_2_ and produces high levels of the plant’s relevant hormone auxin. FP10 is a *Nif*^-^ strain deficient in N_2_-fixation. *ipdC* is a strain that is deficient in auxin production but fixes N_2_. Zn uptake was measured in laboratory-based studies of 3-week-old plants using radioactive ^65^Zn^2+^ (t_½_ 244 days). Principal Component Analysis was applied to draw out correlations between microbial functions and host ^65^Zn^2+^ accumulation. Additionally, statistical correlations were made to our prior data on plant uptake of radioactive ^59^Fe^3+^ and ^59^Fe^2+^. These correlations showed that low microbial auxin-producing capacity resulted in the greatest accumulation of ^65^Zn. Just the opposite effect was noted for ^59^Fe where high microbial auxin-producing capacity resulted in the greatest accumulation of that tracer.

## 1. Introduction

Zinc has been implicated with a broad spectrum of growth characteristics in higher plants. In apple, visible symptoms of Zn deficiency occur in dicotyledons where a significant decrease in leaf size is seen [1]. This trait, coined “little leaf” syndrome, is observed as a common growth characteristic in many fruit tree species subjected to Zn deficiency [2]. Zinc also plays important roles in many biochemical functions within plants. It is an essential component for over 300 enzymes [3]. It also plays a role in DNA and RNA metabolism, cell division, and protein synthesis [4]. A lack of sufficient Zn during plant growth can decrease yield and crop quality because of the disruption in these normal metabolic functions [5,6,7].

Today, approximately 30% of global crop production is lost due to essential nutrient deficiencies caused by climatic extremes that result in excessive soil weathering [8] and lack of diverse agricultural practices that deplete nutrient levels in soil. Additionally, foods produced from Zn-deficient crops can result in human Zn deficiency, which can impact human well-being by reducing immune function and increasing the risk of growth stunting in children or adverse pregnancy outcomes in women [5,9]. 

Plant-growth-promoting bacteria (PGPB), which can help their host weather difficult conditions and potentially increase nutritional value of crops, are finding increased use in agriculture. These organisms can activate physiological and biochemical responses within their host for mutual benefit to build natural tolerances to environmental stresses and thereby reduce losses in the field [10,11,12,13,14,15,16]. Several have been identified as endophytes of grass species, including *Azoarcus* spp. in Kallar grass (*Leptochloa fusca* (L.) Kunth) and rice (*Oryza sativa)* [17,18,19], *Herbaspirillum seropedicae* in sugarcane (*Saccharum officinarum*) [20] and sorghum (*Sorghum bicolor*) [19], and *Gluconacetobacter diazotrophicus* in sugarcane [21]. Others have been identified as epiphytes, including *Azospirillum brasilense* and *lipoferum,* which have been commercialized as crop inoculants for maize and wheat [22,23,24,25]. These strains are gaining increasing acceptance in agriculture as PGPB inoculants. Unlike rhizobia that form an intracellular symbiosis with their legume hosts, PGPB do not induce the formation of observable plant structures (nodules). They are also not usually major components of the soil microflora [20,26]. These N_2_-fixing bacteria infect at the emergence of lateral roots and in the zone of elongation and differentiation above the root tip [14]. Typically, very high numbers of PGPB in roots have been reported (i.e., ≤10^8^ gram^−1^ root dry weight) with no observable disease symptoms [17].

The present work reports on the use of radioactive ^65^Zn (t_½_ 244 days) to examine root assimilation and whole-plant transport of the metal under different conditions of growth. A review of the literature reveals a limited number of papers that have used Zn radioisotopes to examine plant uptake of the metal [26,27,28,29]. Measuring Zn uptake through its radioactive decay can be highly quantitative. However, its general utility in plant biology is limited by the fact that many laboratories are not equipped with the appropriate nuclear instrumentation needed to make such measurements. Here, plants were inoculated with three different functional mutant strains of *Azospirillum brasilense* PGPB, including HM053, a *Nif ^+^* constitutively expressed mutant of the nif gene coding for nitrogen fixation enzymes that fixes excess N_2_ and excretes large amounts of ammonium into the rhizosphere; *ipdC*, a mutant strain disrupted in the ipdC gene thus impaired in biosynthesizing the plant’s relevant hormone auxin, indole-3-acetic acid [16]; and FP10, a *Nif ^–^* mutant that is deficient in fixing N_2_, and also compared plant performance for assimilating ^65^Zn relative to non-inoculated controls. These studies were conducted to determine whether these microbial functions had any influence on their host’s performance. Furthermore, the longer-term effects of microbial functions on host seed filling were examined in outdoor potted plant studies to determine whether harvested kernels had a higher Zn content. 

## 2. Materials and Methods

### 2.1. Bacteria Growth

Functional mutants were grown in liquid NFbHP-lactate medium following published procedures [13]. The concentration of zinc in the growth media was 0.8 μM ZnSO_4_ ⋅7 H_2_O. Cultures were washed with sterile water and diluted to 1 mL containing between 10^6^ to 10^8^ colony-forming units per milliliter (CFU mL^−1^). Bacteria content was measured by sample turbidity, where OD_600_ = 1.0 (optical density at 600 nm, corresponding to 10^8^ cells mL^−1^). Root inoculation involved adding the inoculum to a Petri dish of 10–20 maize seedlings and rocking in a shaking incubator for two hours at 30 °C. Seedlings were then placed into germination pouches for five days before transplanting to hydroponics. Liquid inoculants of each bacterial mutant were made by taking the liquid bacteria cultures described above and centrifuging the cultures down to a pellet. The supernatant above the pellet was removed, and sterile water washes of the pellet were completed for 3 rinses. Upon rinsing the nutrient from the pellet, it was diluted to 1 mL total volume in sterile water and then administered to the plants, both indoors and out. 

### 2.2. Laboratory Plant Growth:

Maize kernels from Elk Mound Seed Co. (Hybrid 8100, Elk Mound, WI, USA) were dark-germinated at room temperature for two days on sterilized paper towels wetted with sterile water in a Petri dish. Seeds were inoculated with bacteria culture as appropriate and transplanted to plastic seed germination pouches (PhytoAB, Inc., San Jose, CA, USA) wetted with sterile Hoagland’s basal salt solution for approximately one week. They were then transferred to individual 600 mL hydroponics cells that were continuously aerated and filled with Hoagland’s nutrient (pH 6.0). The nutrient was exchanged on a five-day cycle. Growth conditions consisted of 12-hour photoperiods, 500 μmol m^−2^ s^−1^ light intensity, and temperatures of 25 °C/20 °C (light/dark) with humidity at 70–80%. 

### 2.3. Outdoor Plant Growth

For outdoor, non-radioactive studies, 3 maize kernels from Elk Mound Seed Co. (Hybrid 8100) were sown into each of 2.7-gallon pots filled with ProMix. Plants were placed on elevated tables outside and pots were rotated every week to ensure uniformity of growth conditions. After germination, any excess seedlings were removed from each pot leaving a single plant. A capful of fertilizer (~1.2 g) containing nitrogen, phosphate, and potash (14-14-14, Osmocote™ Smart-Release Plant Food Flower & Vegetable™, The Scotts Company, Marysville, OH, USA) was added to the assigned pots at the time of planting. Fertilizer was reapplied to pots 30 and 60 days after germination (DAG). Study regimes included the following: (i) non-inoculated control plants; (ii) plants inoculated with *A. brasilense* HM053 bacterium; (iii) plants inoculated with *A. brasilense ipdC* bacterium; and (iv) plants inoculated with *A. brasilense* FP10 bacterium. Plants were administered liquid inoculants at 21, 42, and 63 DAG using re-washed bacteria cultures containing between 10^6^ to 10^8^ colony-forming units per milliliter, as described above. These cultures were further diluted to 10 mL volumes in sterile water and administered to the pots. Treatments were randomized across the planting platforms. At the end of the growing season, cobs were harvested, and seeds analyzed by ion chromatography for Zn content.

### 2.4. ^65^Zinc Studies

^65^Zn was purchased from PerkinElmer Life Sciences (Billerica, MA USA). One hour before administration of radiotracer, plants were removed from their hydroponics cells and suspended in 600 mL beakers consisting of 100 mL of deionized water (Figure 1). Plants were maintained at the same daytime light and temperature conditions as that used to maintain their growth. An aqueous solution of ^65^Zn radiotracer at 0.74 MBq was injected into the beaker of water in which the roots were immersed. Based on the radiotracer’s specific activity, we estimated that 45 μg of non-radioactive Zn was introduced to the 100 mL of deionized water during a tracer study (equivalent to 0.7 μM), which closely matched the 0.8 μM Zn levels introduced via the Hoagland’s nutrient solution during plant growth. Hence, the mass of Zn introduced in the tracer studies did not perturb the plant’s normal exposure to this micronutrient. A radiation detector (Eckler & Ziegler, Inc., Berlin, Germany 1-inch Na-PMT, photomultiplier tube gamma radiation detector) affixed to the plant 8 cm above the base of the stem provided dynamic feedback on ^65^Zn transport from roots to shoots. Data were acquired at a 1 Hz sampling rate using 0-1 V analog input into an acquisition box (SRI, Inc, Torrance, CA, USA). After 3 h of acquisition, roots were cut from the shoots, thoroughly washed in water, blotted dry, and weighed. Shoots were also weighed. Both root and shoot tissues were then sequentially placed in a 3 inch NaI-PMT gamma well-type detector for quantifying the amount of ^65^Zn radioactivity. ^65^Zn uptake and allocation percentages were calculated as the amount of radiotracer counted in the plant roots and shoots divided by total radioactivity administered as a percentage and the amount of radioactivity measured solely in the shoots divided by the total radioactivity in the roots and shoots as a percentage, respectively.

### 2.5. Plant Radiography

After ^65^Zn administration, plants were harvested and roots were blotted dry and laid out on an absorbent pad for imaging. Shoots were also laid out on a separate absorbent pad for imaging. Radiographic images of different tissue areas (roots and shoots) were acquired by exposing phosphor plate films. Phosphor plates of roots were exposed for 36 h while plates of shoots were exposed for 120 h to acquire a sufficient signal. After exposure, phosphor plates were then read using the Typhoon 9000 imager (Typhoon^TM^ FLA 9000, GE Healthcare, Piscataway, NJ, USA). Images were only used qualitatively for determining spatial patterning of ^65^Zn tracer in roots and shoots; hence no attempt was made to normalize image data. Comparative whole-plant radiographic images of ^59^Fe^3+^ and ^59^Fe^2+^ were also acquired from our prior work [16], but because of the faster decay rate of this radionuclide (t_½_ 44.5 day), we only needed to expose these tissues for 16 h. 

### 2.6. Ion Chromatography Analysis of Corn Kernel Zn Content

Zinc content was quantified from corn kernels using ion chromatography coupled with UV absorption detection following the collection and drying of the kernels in an oven for 3 weeks at 65 °C. Seeds were pulverized between plastic sheets using a wooden mallet and dissolved in 1 mL of 1M HCl. Samples were subjected to ultrasonication for 5 minutes at 100% amplitude (Branson Bransonic 32; Sigma-Aldrich Corp. St. Louis, MO, USA) then centrifuged for 15 minutes at 3000 rpm. The supernatant was removed for sampling and stored in brown glass vialsin a refrigerator (2–8 °C). Zn standards were prepared in 0.1 M HCl using zinc chloride (ZnCl_2_, 1 mg mL^−1^).

The analytical system consisted of a Thermo Scientific Dionex AXP Metal-Free HPLC with a Rheodyne metal-free injector and PEEK tubing 1/20 cm inner diameter. The ion chromatography column was a Thermo Fisher Scientific™ Dionex™ (Waltham, MA, USA) IonPac CS5A 4 i.d. × 250 mm analytical column outfitted with a CG5A 4 i.d. × 40 mm guard column designed to separate a broad range of metal complexes by cation and anion chromatography. The mobile phase consisted of 7 mM pyridine-2,6-dicarboxylic acid, 66 mM potassium sulfate, and 74 mM formic acid pH 4.2 run at a flow rate of 1.2 mL min^−1^. A post column reagent comprising 0.5 mM 4-(2-pyridylazo) resorcinol (Dionex Corp., Sunnyvale, CA, USA) in MetPac PAR post column diluent (1.0 M 2-dimethylaminoethanol/0.50 M ammonium hydroxide/0.30 M sodium bicarbonate pH 10.4) at a flow rate of approximately 0.6 mL min^−1^ was used for detection by a Knauer Smartline 2500 UV detector operated at 530 nm. Sterile water (HyPure™ WFI Quality Water, HyClone Laboratories, Logan, UT, USA) was used in solvent preparation. All biological samples were analyzed in triplicate. 

### 2.7. Statistical Analysis

Data were subjected to the Shapiro–Wilk Normality Test to identify outliers, so all data groups reflected normal distributions. Data were analyzed using the Student’s *t*-test for pair-wise comparisons made between non-inoculated controls and bacteria treatment. Statistical significance was set at *p* < 0.05. 

### 2.8. Principal Component Analysis of ^65^Zn and ^59^Fe Data

The ^65^Zn uptake and allocation data from the present study and ^59^Fe data from our prior work [16] were analyzed by Principal Component Analysis (PCA) using XLSTAT software version 2020.3 (Addinsoft Inc., New York, NY 10001, USA). 

## 3. Results and Discussion

Results in Figure 2 of the different rates for ^65^Zn transport as a function of *A. brasilense* inoculation showed that *ipdC* > HM053 > FP10. FP10 was most like non-inoculated control plants. Tissue distribution of Zn using ‘cut and count’ techniques revealed a similar dissimilarity between *ipdC* bacteria and the other inoculants (Figure 3). Systematic trends defining uptake and in *planta* translocation of ^65^Zn become apparent in the PCA biplot (Figure 4A). The information included in the biplot was represented by two principal components (PC), with PC1 representing 71.89% of the information embedded in the data and PC2 representing 28.11%. The PCs selected to represent the data are classified as feature vectors (F1 and F2), as shown on the biplot. The axes are in terms of the eigenvalues, with larger values indicating a greater variance and thus a greater representation of the information within the data. The active variables, shown as dotted lines, represent the initial variables of root assimilation of ^65^Zn and shoot allocation. The length of the active variable vectors indicates how well the variables are tied to the feature vectors. Since the active variable vectors are equivalent in length in Figure 4A and are found equally between F1 and F2, it can be interpreted that both active variables are equally represented by both F1 and F2. As displayed, each of the microbial treatments clustered together, indicating behavior within a treatment type that was distinct from other treatments. It was shown that FP10 and non-inoculated maize were similar in overall ^65^Zn uptake and shoot allocation behavior. HM053 inoculated maize exhibited a slight elevation in allocation patterns relative to control and FP10. *ipdC* was most unique in its uptake and allocation patterns than other treatments in the X- and *Y*-axis directions. 

What distinguishes *ipdC* from the other microbial inoculants examined in this study is its deficiency in producing auxin (indole-3-acetic acid), an important plant hormone. Our past studies showed that the HM053 mutant exhibited the highest level of auxin biosynthesis, being 2 times that of FP10 and 13 times that of *ipdC* [16]. We know that auxin biosynthesis in plants and Zn levels are strongly correlated [30,31,32]. With tryptophan being the principal intermediate in auxin biosynthesis, withholding Zn was shown to lower plant tryptophan levels [30] and auxin levels [31], while exogenous treatment with Zn increased tryptophan levels [32]. We suspect that the mechanism for promoting plant ^65^Zn uptake in the present study has to do with the auxin-producing capacity of the microorganism. We note that while *ipdC* lacks the ability to biosynthesize auxin, it still processes the molecular machinery to produce indole—a key precursor to tryptophan biosynthesis [16]. In fact, maize root indole emissions with *ipdC* inoculation were nearly 2 times that of HM053 inoculated plants, and 1.5 times that of FP10 inoculated plants [16]. We suspect this behavior may be due to bacteria-root indole trafficking, which could elevate the endogenous pool of plant tryptophan, causing an elevation in Zn uptake. To the best of our knowledge, no one has examined whether tryptophan treatments will elevate endogenous levels of plant Zn. 

Similar statistical treatments were applied to our previously published ^59^Fe data [16], both for ferrous (Fe^2+^) and ferric (Fe^3+^) forms of the tracer to yield Figure 4B,C, respectively. As displayed here, each of the microbial treatments again clustered together, indicating the behavior within a treatment type that was distinct from the other microbial treatments. Unlike our ^65^Zn^2+^ data shown in Figure 4A, we observed different microbial influences on host iron assimilation, with HM053 exhibiting the greatest influence for root uptake and shoot allocation of both ^59^Fe^3+^ and ^59^Fe^2+^ over the other bacteria strains. In our earlier work, we ascertained through whole-plant radiographic imaging that the oxidation state of the iron radiotracer was unaltered by the microorganism’s functions. Here, we noted that each oxidation state of the radiometal exhibited a different spatial patterning across the shoot tissues with ^59^Fe^3+^ accumulating in leaf tips, while ^59^Fe^2+^ accumulating uniformly throughout the leaves. Figure 5 shows an example of this distribution from HM053 inoculated maize plants since HM053 caused the largest increase in ^59^Fe^3+^ and ^59^Fe^2+^ allocations to shoots relative to the other microbial inoculants. For comparison, we also show radiographic images in the same figure from maize ^65^Zn^2+^ studies as a function of *ipdC*, FP10, and HM053 microbial inoculants. In all cases here, ^65^Zn spatial patterning in leaves was similar to that of ^59^Fe^2+^. However, root tissues exhibited significantly different radiotracer distributions, where elongation zones showed higher levels of ^65^Zn than both oxidation states of the ^59^Fe radiotracer. Additionally, we noted a common trend where a high accumulation of ^65^Zn was observed in the lower stem region, likely in the coleoptile. Past studies have shown that the coleoptile in maize seedlings exhibits a strong growth dependency on auxin [33]. Taken together, our results show that maize assimilation of divalent metals such as Zn^2+^ or Fe^2+^ has significant dependency on microbial auxin biosynthesis. Once assimilated, these metals also exhibit very different spatial patterning during transport aboveground.

Finally, we examined the longer-term influence of these mutant strains of *A. brasilense* on kernel zinc content. Results in Figure 6 showed that HM053 did not alter seed zinc levels relative to non-inoculated controls. However, both *ipdC* and FP10 bacteria showed significantly less zinc content. Hence, while *ipdC* promotes zinc accumulation in host vegetative tissues, that action does not translate to the seed filling process. We suspect that heavy ^65^Zn accumulation in the lower stem may minimize the metal’s availability during seed filling.

## 4. Conclusions

The present work showed evidence that certain biological traits of root-associating microorganisms can have beneficial effects on the host plant in promoting Zn uptake. These mechanisms of action appeared to correlate with the auxin producing capacity of the microorganism in that the auxin deficient mutant *ipdC* had the greatest influence in promoting host ^65^Zn uptake. Mechanisms of action appear not to be universally translated across all metals, since ^65^Zn and ^59^Fe exhibited very different dependencies on microbial functions.

While the microbial actions promoting host Zn uptake could benefit over the long term in improved crop yield, our results suggest that there is little or no effect on kernel Zn content improving its nutritional value.

## Figures and Tables

**Figure 1 microorganisms-09-01002-f001:**
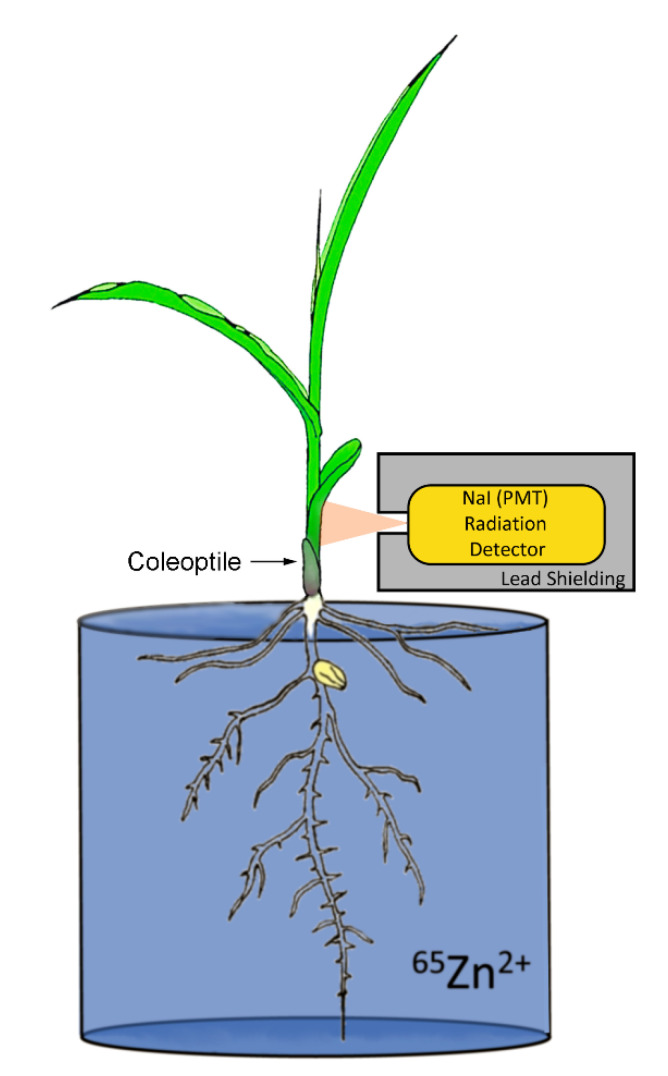
Experimental setup used for plant ^65^Zn uptake studies. During exposure to radiotracer, plants were maintained at a constant 500 μmol m^−2^ s^−1^ light intensity and 21 °C temperature.

**Figure 2 microorganisms-09-01002-f002:**
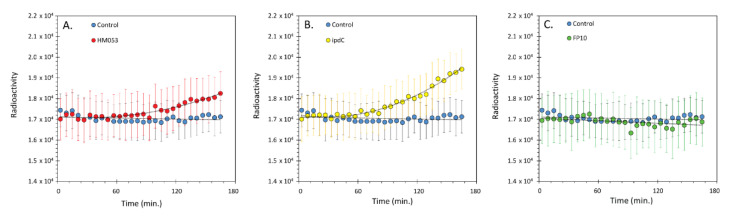
Dynamic ^65^Zn transport over ~3-hours of acquisition. The data reflect ^65^Zn transport measured by a radiation detector located along the plant stem at approximately 8 cm from the base of the stem. Each data point represents a 5-minute average ± SD of 300 data points sampled at 1 Hz across 5-6 biological replicates. (Panel **A**): Transport of ^65^Zn in HM053-inoculated plants is shown in red with the non-inoculated control data shown in black. (Panel **B**): Transport of ^65^Zn in *ipdC* inoculated plants is shown in yellow, with the non-inoculated control data shown in black. (Panel **C**): Transport of ^65^Zn in FP10 inoculated plants is shown in green, with the non-inoculated control data shown in black. In all cases, data were normalized to the same starting level of radioactivity for direct comparison of transport across the bacteria strains. Data were fitted to trendlines (depicted by the dashed lines).

**Figure 3 microorganisms-09-01002-f003:**
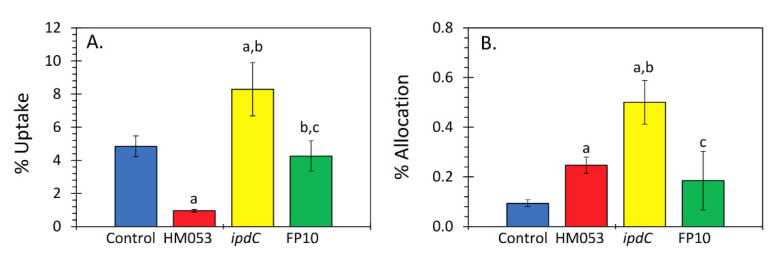
(Panel **A**): ‘Cut and count’ measurements yielded information on plant uptake of ^65^Zn presented as the percentage of tracer dose administered to the beaker that was assimilated by the plant over 3 h. (Panel **B**): Root-to-shoot allocation of ^65^Zn is presented as the percentage of the administered dose of radiotracer. Data reflect means for *N* = 5-6 replicates (±SE). Statistical significance *p* < 0.05 was designated by ‘a’ in a comparison of treatment to control, ‘b’ in a comparison of *ipdC* or FP10 to HM053, and ‘c’ in a comparison of FP10 to *ipdC* treatment.

**Figure 4 microorganisms-09-01002-f004:**
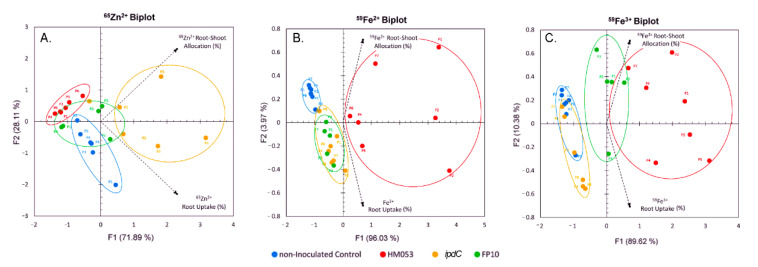
Principal Component Analysis correlates ^65^Zn uptake and root–shoot allocation (Panel **A**) to the biological functions of the beneficial microbes. HM053 and FP10 mutant strains were most like non-inoculated plant behavior, while *ipdC* was dissimilar, exhibiting the highest levels of root uptake and transport. Statistical correlations were made to ^59^Fe^2+^ (Panel **B**) and ^59^Fe^3+^ data that we acquired in prior work [16].

**Figure 5 microorganisms-09-01002-f005:**
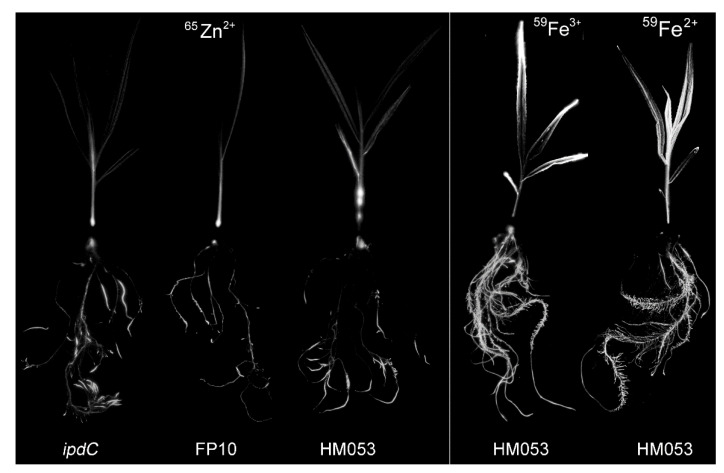
Radiographic images of maize plants after exposure to ^65^Zn as a function of inoculation with *ipdC*, FP10, and HM053 bacteria. For comparison, we also show radiographic images of maize plants after exposure to ^59^Fe^3+^ and ^59^Fe^2+^ radiotracers from our prior work [16]. These plants were also inoculated with HM053, which was shown to exert the greatest influence on host ^59^Fe^3+^ and ^59^Fe^2+^ uptake and allocation.

**Figure 6 microorganisms-09-01002-f006:**
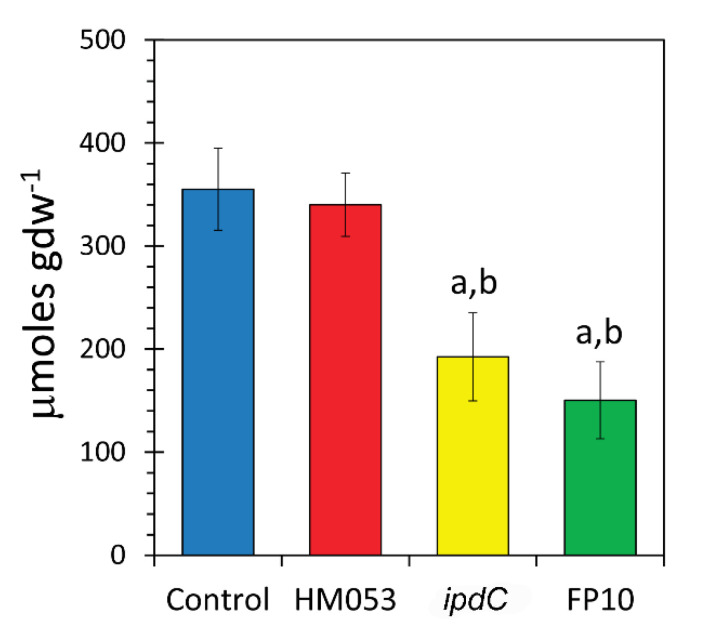
Kernel Zn content was measured using ion chromatography. Data reflect means for *N* = 12 replicates (±SE). Statistical significance *p* < 0.05 was designated by ‘a’ in the comparison of treatment type to untreated control, and ‘b’ in a comparison of *ipdC* or FP10 treatments to HM053 treatment.

## Data Availability

The data presented in this study are available upon requests made to the corresponding author and pending institutional permissions.

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
