# Peer review of "Plant-Growth-Promoting Bacteria Can Impact Zinc Uptake in Zea mays: An Examination of the Mechanisms of Action Using Functional Mutants of Azospirillum brasilense"

_microorganisms, 2021, doi:10.3390/microorganisms9051002_

Round 1
Reviewer 1 Report
This manuscript describes how the inoculation of Azospirillum brasilense functional mutants HM053, ipdC, and FP10 affect Zn uptake in host maize plants This study utilized the radioactive Zn-65 tracer to estimate Zn uptake from the root and allocation to the shoot using non-invasive radiation monitoring method and destructive radiation measurement methods and revealed that ipdC mutant had the greatest influence in promoting Zn uptake of host plant. Although kernel Zn content in the ipdC-inoculated plants was not increased in the outdoor experiment, this study will provide useful information for research on enhancing nutrients uptake using microorganisms.
The comments to make this manuscript more understandable are as follows.
1) In Figure 2, the signal and noise cannot be distinguishable because the data is plotted every one second. It is recommended that the average values per 10 minutes be plotted with error bars for the standard deviation of the biological replicates. Also, it should be divided into four plots, A(Control), B(HM053), C(ipdC), and D(FP10).
2) In Figure 3, It is not obvious how to calculate the amount of "Uptake" and "Allocation". The amount of "Uptake" is the sum of radioactivity in the roots and shoot divided by the administered dose? The amount of "Allocation" is radioactivity in the shoot divided by the administered dose (or radioactivity in the root)? The details of the calculation method should be provided in the section "2.4. 65Zinc Studies".
3) In L108, the description of "liquid inoculants" was incomprehensible. It should describe the method of inoculation of the bacteria in the outdoor experiment.
4) In L127, it should be stated whether or not non-radioactive zinc was added with Zn-65.
Author Response
Aside from the fact that Reviewer 1 checked the box indicating that our manuscript required “extensive editing of English language and style” which I believe was mistakenly checked, we did make improvements to the Methods for greater clarity and readability. Here, we revised the order of many of the subsections within the Methods providing a clearer flow of information for the reader. Additional detail on how the studies were conducted was also added in the Methods, as per specific comments below.
Specific Reviewer items addressed below with Author responses italicized:
- In Figure 2, the signal and noise cannot be distinguishable because the data is plotted every one second. It is recommended that the average values per 10 minutes be plotted with error bars for the standard deviation of the biological replicates. Also, it should be divided into four plots, A(Control), B(HM053), C(ipdC), and D(FP10). We revised Figure 2 averaging time-activity data across 5-minute intervals rather than the 10-minute interval as suggested by the reviewer and plotted the data with error bars designating SD. The figure caption was revised to reflect this treatment of the data. However, we still maintained the 3-panel format showing the 65Zn transport data from each bacteria strain as compared with non-inoculated control data.
- In Figure 3, It is not obvious how to calculate the amount of "Uptake" and "Allocation". The amount of "Uptake" is the sum of radioactivity in the roots and shoot divided by the administered dose? The amount of "Allocation" is radioactivity in the shoot divided by the administered dose (or radioactivity in the root)? The details of the calculation method should be provided in the section "2.4. 65Zinc Studies". We added additional detail (see lines 151-155) on how uptake and allocation were calculated.
- In L108, the description of "liquid inoculants" was incomprehensible. It should describe the method of inoculation of the bacteria in the outdoor experiment. We added additional detail (see line 93-100) on bacteria harvesting and text (see lines 123-125) on how microbial treatments were introduced to the potted plants in the outdoor studies.
- In L127, it should be stated whether or not non-radioactive zinc was added with Zn-65. No non-radioactive zinc was intentionally added during our studies. However, there was zinc mass associated with the radiotracer. We calculated 45 μg of zinc was introduced to each plant based on the specific activity of the radiotracer which would be equivalent to 0.7 μM Zn for the 100 mL of water used to expose roots to 65Zn We noted that Hoagland’s nutrient contained 0.74 μM Zn. Hence the tracer studies were conducted at about the same level of physiological Zn the plant experienced during its growth. This has been detailed on lines 144-149.
Reviewer 2 Report
The manuscript entitled “Plant Growth Promoting Bacteria Can Impact Zinc Uptake in Zea mays: An Examination of the Mechanisms of Action using Functional Mutants of Azospirillum brasilense,.” by Housh et al. given experimental evidence that zinc absorption by the plant using bacterial mutants. It deserves merits of publication, however, I have some comments.
Minor corrections
- In the abstract, remove the few initial lines because they resemble the introduction of course they are repeated in the introduction part also.
- Line 57-61----the provided all references are only increasing Zn? Or other biochemical processes also. If so, rewrite the sentence.
- Since the Zn content is not accumulated in Kernal, the portion of nutritional importance to humans may be removed from this manuscript.
- Figure 2 legend is not informative; all figure parts need to explain in the legend.
- The authors need to provide a clear explanation that should disclose the correlation between tryptophan and Zn uptake.
- One more basic confusion is that why authors selected iron mutants that are deficient in auxin synthesis to prove the Zn uptake. Is there any interlink please make clear.
Specific comments
- Line 60 Azospirillum lipoferum should be written as lipoferum
- Line 90 - Petri should be written as Petri and throughout the manuscript
- What was the Zinc content in the Hoagland solution? need mention in materials and methods
- Line 261- italics brasilens.
Hence the overall recommendation is to address the minor suggestions in the manuscript with moderate revisions that meet the required suggestions mentioned above.
Author Response
1. In the abstract, remove the few initial lines because they resemble the introduction of course they are repeated in the introduction part also. The Abstract has been revised and shortened to remove redundancy.
2. Line 57-61----the provided all references are only increasing Zn? Or other biochemical processes also. If so, rewrite the sentence. The paragraph within the Introduction from which lines 57-61 of the original submission are referred to “These organisms can activate physiological and biochemical responses within their host for mutual benefit to build natural tolerances to environmental stresses and thereby reduce losses in the field [10-16],” is a general statement on PGPB and how they benefit plant growth. References 10-16 are not specific to increasing Zn as the Reviewer was asking. However, any manuscript addressing PGPB would be negligent if these general background references were not included in the Introductory text about them.
3. Since the Zn content is not accumulated in kernel, the portion of nutritional importance to humans may be removed from this manuscript. We decided to leave the few lines addressing Zn nutrition and human health in the Introduction (see lines 49-51) as it was a driver for why we initiated these plant-microbe studies to determine whether improved host Zn accumulation transfers to seed filling and improved crop nutrition. As it turns out, we did not observe an improvement in corn kernel Zn content.
4. Figure 2 legend is not informative; all figure parts need to explain in the legend. The revised manuscript now includes a revised Figure 2 as per suggestions from Reviewer 1. The new presentation of the data in Figure 2 has greater clarity. The figure caption was also revised to reflect greater detail on our averaging data points over 5-minute intervals.
5. The authors need to provide a clear explanation that should disclose the correlation between tryptophan and Zn uptake. Perhaps the reviewer missed this section but in lines 253-269 we wrote in an extensive analysis of our results drawing from the literature on what is known about tryptophan and plant Zn. From that section we wrote: “What distinguishes ipdC from the other microbial inoculants examined in this study is its deficiency in producing auxin (indole-3-acetic acid), an important plant hormone. Our past studies showed that the HM053 mutant exhibited the highest level of auxin biosynthesis, being 2-times that of FP10 and 13-times that of ipdC [16]. We know that auxin biosynthesis in plants and Zn levels are strongly correlated [30-32]. With tryptophan being the principal intermediate in auxin biosynthesis, withholding Zn was shown to lower plant tryptophan levels [30], and auxin levels [31], while exogenous treatment with Zn increased tryptophan levels [32]. We suspect that the mechanism for promoting plant 65Zn uptake in the present study has to do with the auxin producing capacity of the microorganism. We note that while ipdC lacks the ability to biosynthesize auxin, it still processes the molecular machinery to produce indole – a key precursor to tryptophan biosynthesis [16]. In fact, maize root indole emissions with ipdC inoculation were nearly 2-times that of HM053 inoculated plants, and 1.5-times that of FP10 inoculated plants [16]. We suspect this behavior may be due to bacteria-root indole trafficking which could elevate the endogenous pool of plant tryptophan causing an elevation in Zn uptake. To the best of our knowledge, no one has examined whether tryptophan treatments will elevate endogenous levels of plant Zn.”
6. One more basic confusion is that why authors selected iron mutants that are deficient in auxin synthesis to prove the Zn uptake. Is there any interlink please make clear. We choice to include the iron data to demonstrate that metal uptake and transport in plant associating with these bacteria is not universal. That is Zn2+ uptake and transport is very different from Fe2+ uptake and transport. It is also important to realize that the interactions of the bacteria with their host for promoting metal nutrient uptake is very different for two different elements of the same divalent oxidation state.
7. Line 60 Azospirillum lipoferum should be written as lipoferum Correction was made.
8. Line 90 - Petri should be written as Petri and throughout the manuscript Corrections were made.
9. What was the Zinc content in the Hoagland solution? need mention in materials and methods. The Zn content from the Hoagland’s nutrient was detailed in lines 144-149 where we discuss the Zn mass contribution from the 65Zn radiotracer.
10. Line 261- italicize brasilense. Corrections were made.